# Beryllium Dimer Reactions with Acetonitrile: Formation of Strong Be−Be Bonds

**DOI:** 10.3390/molecules29010177

**Published:** 2023-12-28

**Authors:** Fei Cong, Liyan Cai, Juanjuan Cheng, Zhen Pu, Xuefeng Wang

**Affiliations:** 1Shanghai Key Lab of Chemical Assessment and Sustainability, School of Chemical Science and Engineering, Tongji University, Shanghai 200092, China; 1810918@tongji.edu.cn (F.C.); 2110184@tongji.edu.cn (L.C.); chengjuanjuan@tongji.edu.cn (J.C.); 2China Academy of Engineering and Physics, Mianyang 621900, China

**Keywords:** Be−Be bonds, beryllium dimer, infrared spectrum, quantum chemical calculations

## Abstract

Laser ablated Be atoms have been reacted with acetonitrile molecules in 4 K solid neon matrix. The diberyllium products BeBeNCCH_3_ and CNBeBeCH_3_ have been identified by D and ^13^C isotopic substitutions and quantum chemical calculations. The stabilization of the diberyllium species is rationalized from the formation of the real Be−Be single bonds with bond distances as 2.077 and 2.058 Å and binding energies as −27.1 and −77.2 kcal/mol calculated at CCSD (T)/aug-cc-pVTZ level of theory for BeBeNCCH_3_ and CNBeBeCH_3_, respectively. EDA-NOCV analysis described the interaction between Be_2_ and NC···CH_3_ fragments as Lewis “acid−base” interactions. In the complexes, the Be_2_ moiety carries positive charges which transfer from antibonding orbital of Be_2_ to the bonding fragments significantly strengthen the Be−Be bonds that are corroborated by AIM, LOL and NBO analyses. In addition, mono beryllium products BeNCCH_3_, CNBeCH_3_, HBeCH_2_CN and HBeNCCH_2_ have also been observed in our experiments.

## 1. Introduction

The chemistry of beryllium is predicted to be the richest among the alkaline earth metals due to its small size and the highest electronegativity and ionization energy among *s*-block elements [1,2], which is, however, largely unexplored due to the high toxicity of the compounds containing this element [3,4]. Over the past few decades, the very weak beryllium–beryllium interaction has been the most studied for *s*-block metal–metal interactions [5,6,7]. Isolated beryllium dimer was characterized as a typical weakly bound molecule with a bonding interaction of 11.2 kJ mol^−1^ (calc.) and a large bond distance of 2.45 Å as a result of the both doubly occupied bonding 2σ_g_^+^ orbital and antibonding 2σ_u_^+^ orbital [8,9,10].

Up to now, diberyllium complexes have aroused a great deal of enthusiasm among researchers, and many theoretical strategies have been proposed to enhance the strength of Be−Be bond [5,11,12,13,14,15]. Removing/adding an electron from/to this dimer by interaction with some other molecules is a general formula to stabilize the system. The complexation of Be_2_ with electron-withdrawing ligands such as F [16] or CN [17], or electron-deficient conjugated fragments such as cyclopentdienyl [18,19], pentadienyl [20] and phospholyl [21] is able to obtain a classical Be−Be *σ* single bond by pulling one or both electrons from antibonding orbital of Be_2_ to ligands. Despite the intrinsic electron-deficient nature of beryllium, reports indicate that the Be_2_ moiety in these molecules is essentially a dication Be_2_^2+^, [17] which exhibits the formation of a strong bond in the presence of a counterion. Strong Be–Be bonds are also formed via electron transferring from Be atoms to C_n_H_n_ (n = 3, 5, 7), *π*-radicals [22] and SO ligands. [23] Furthermore, a neutral odd-electron Be−Be bond is firstly identified in the tri-AMD-ligands-chelated D_3h_-Be_2_(AMD)_3_ complex by one AMD ligand attracting one electron from Be antibonding σ* orbital [24]. A strong 2-center-1-electron Be−Be bond is also formed by attaching one electron to 1,2-diBeX-benzene (X = H, F, Cl, CN) derivatives [25]. Be–Be double-π bonds are first achieved in the predicted octahedral cluster of Be_2_(μ_2_-X)_4_ (X = Li, Cu, BeF) by a novel concerted electron-donation from four *s*^1^-type electron-donating ligands [15]. The Be–Be triple bonds are formed in theoretically predicted Li_6_Be_2_ [26] species, which are stabilized by six *s*^1^-type donor ligands and in Be_2_X_4_Y_2_ (X = Li, Na; Y = Na, K) [27] clusters where six alkali metals are electron-donating ligands. Another way to form a Be−Be covalent bond is by adding radical ligands to an excited Be_2_ moiety. The neutral Be_2_ moiety in an excited state adding ligands such as (HCNMe)_2_B [28] and *N*-heterocyclic carbenes (NHCs) [29] can form a single or double bond, respectively.

Acetonitrile is known as an effective electron donor based on the previous research of reactions with metal atoms [30,31,32,33,34,35,36,37], and it could be a good candidate for stabilizing the Be_2_ dimer. In this paper, we investigated the reactions of beryllium atoms with acetonitrile by means of matrix-isolation infrared spectroscopy and theoretical calculations, in order to further supplement the reactions of alkali metal with acetonitrile, and search for stable diberyllium complexes that might be formed. Six different products were spectroscopically identified in solid neon including two complexes possessing Be−Be single bonds, and three related reaction paths were presented.

## 2. Results and Discussion

The infrared spectra for acetonitrile (0.5% in solid neon) on pre-deposition and the reaction products on co-deposition, annealing and stepwise photolysis in the selected regions are shown in Figure 1. The isotopic substitution experiments with CD_3_CN and ^13^CH_3_^13^CN samples were performed and the infrared spectra are given in Figure 2. The absorptions of acetonitrile are extremely strong, so the changes of acetonitrile absorptions during the reaction are negligible compared with the products’ absorptions. In addition to the absorptions assigned to the precursor and common species that were also observed in other experiments with CH_3_CN as reagent, several new product absorptions appeared, which can be classified into six groups based on their intensity changes. Table 1 shows the observed and computed vibrational frequencies of the products along with the assignment of the modes. The full sets of B3LYP calculated harmonic and anharmonic frequencies are all collected in Appendix A. The laser intensity is relatively high in our experiments that are able to directly produce beryllium dimer. The infrared intensities of diberyllium products are much weaker with low laser power, while the intensities of mono beryllium products almost remain unchanged in Figure 1a. Several new absorptions located in N−C and C−N stretching region are almost diminished with low laser intensity. We tentatively assigned these weak absorptions to Be_x_(CH_3_CN), which were generated from the reactions of Be clusters and acetonitrile with high laser power.

The relatively weak absorptions in the original deposition spectra marked “**1**” were completely destroyed under visible irradiation. New product absorptions denoted with “**2**” also appeared upon sample deposition, which decreased after annealing to 8 K and totally disappeared on the first visible (λ = 520 nm) irradiation. On the opposite, the intensities of the bands marked with “**3**” and “**4**” tripled and doubled upon full arc irradiation (λ > 220 nm) compared with the co-deposition spectra. In addition, the relatively weak absorptions marked “**5**” and “**6**” only appeared after full arc irradiation.

### 2.1. End-On Products: BeBeNCCH_3_ and BeNCCH_3_

The absorption of C−N stretching mode for BeBeNCCH_3_ complex was observed at 1927.8 cm^−1^ on co-deposition, and its corresponding ^13^C counterpart was located at 1889.4 cm^−1^, but unfortunately the corresponding D isotopic absorption was covered by the precursor bands. The observed absorptions matched well with the computed IR frequencies with strongest intensities at 1986.0 and 1942.7 cm^−1^ for ^12^C and ^13^C, respectively. The band at 994.1 cm^−1^ is attributed to the CH_3_ bending mode, with D and ^13^C counterparts observed at 827.0 and 981.2 cm^−1^, respectively. The computed infrared absorptions of BeBeNCCH_3_ are shown in Appendix A.

The assignment of the mono beryllium end-on complex BeNCCH_3_ was confirmed by the experimental observation of three bands at 1915.4, 1082.3 and 831.4 cm^−1^. The strong 1915.4 cm^−1^ band showed a large ^13^C shift to 1875.4 cm^−1^, exhibiting a ^12^C/^13^C isotopic frequency ratio of 1.0213, while its corresponding D counterpart was at 1911.2 cm^−1^. We assigned this band to the C−N stretching mode of BeNCCH_3_ based on the isotopic shifts and good consistence with the calculated value of 1953.5 cm^−1^. The bands at 1082.3, 1020.8 and 1068.7 cm^−1^ for ^12^C, D and ^13^C isotopes are in good agreement with the calculated results of N−Be stretching mode at 1106.0 (^12^C), 1043.4 (D) and 1091.5 cm^−1^ (^13^C), respectively. The absorption observed at 831.4 cm^−1^ with a ^13^C isotopic substitution at 811.6 cm^−1^ lays in the region expected for a C−C stretching vibration, [37] and the large ^12^C/^13^C ratio of 1.0244 was obtained to verify our assignment. The corresponding D isotopic absorption was covered by the precursor bands. All calculated frequencies of BeNCCH_3_ are listed in Appendix A.

### 2.2. Insertion Products: CNBeBeCH_3_ and CNBeCH_3_

The strong band at 2111.0 cm^−1^ is due to the N−C stretching vibration for the diberyllium insertion product CNBeBeCH_3_, and the D and ^13^C isotopic substitutions were observed at 2111.9 and 2068.1 cm^−1^, respectively, conforming to the N−C stretching mode shifts. The low-intensity band of CH_3_ wagging mode is located at 1226.1 cm^−1^, the predicted value of which is at 1249.1 cm^−1^. The most decisive band for the diberyllium insertion product was observed at 1105.6 cm^−1^, which matched well with the calculated absorption for Be−Be stretching mode at 1109.8 cm^−1^. Unfortunately, the isotopic counterparts for these two absorptions were too weak to be observed in our experiments. The C−Be stretching mode was observed at 1013.4 and 913.4 cm^−1^ in D and ^13^C spectra, but this mode was covered by the extremely strong precursor absorption in ^12^C experiments.

The most intense absorption at 2100.2 cm^−1^ in the spectrum after full arc irradiation showed essentially no shift upon D substitution, but largely shifted to 2064.3 cm^−1^ on ^13^C. The B3LYP isotope ratio for ^13^C substitution of 1.0187 was in accordance with the experimental ratio of 1.0174. We therefore assigned the uniquely strong 2100.2 cm^−1^ band to the N−C stretching vibration of the mono beryllium insertion complex. The strong CH_3_ wagging absorption observed at 1237.8 cm^−1^ showed its D and ^13^C isotopic absorptions at 1205.0 and 1223.8 cm^−1^, and these bands are in excellent consistence with calculated values of 1269.2, 1217.9 and 1255.5 cm^−1^, respectively. A medium absorption was observed at 1162.9 cm^−1^ along with its ^13^C counterpart at 1160.5 cm^−1^, and it is indicative of the antisymmetric CBeN stretching vibration. The observed bands are in the good agreement with the calculated frequencies of 1186.4 and 1183.2 cm^−1^ for ^12^C and ^13^C isotopes. Though the band at 668.3 cm^−1^ is common in the matrix spectra of CH_3_CN + M reactions, it only doubled under full arc irradiation in the case of beryllium, and tracked with other bands labeled “**4**”. Unfortunately, its D and ^13^C counterparts were not detected because of our detector noise. We tentatively assigned this band to the CBeN bending mode of CNBeCH_3_. The B3LYP functional predicted this band at 695.6 cm^−1^, which is only 27.3 cm^−1^ higher than observed. The other infrared absorptions of CNBeCH_3_ are very weak, which are not observed in the experiment (Appendix A).

### 2.3. HBeCH_2_CN and HBeNCCH_2_

The band at 2141.4 cm^−1^ only appeared after full arc irradiation showing almost no ^13^C shift at 2141.8 cm^−1^. The Be−D stretching mode of HBeCH_2_CN was calculated at 1654.6 cm^−1^, and was observed at 1608.3 cm^−1^ among the strong absorptions of water impurity existing in all our experiments. We assigned these bands to the Be−H stretching mode of HBeCH_2_CN. In the low frequency region, a weak absorption at 685.9 cm^−1^ tracked with 2141.1 cm^−1^ band, and it showed a large D shift to 579.9 cm^−1^ and very small ^13^C shift to 680.6 cm^−1^. The band position and large D shift of 106.0 cm^−1^ suggested a Be−H bending vibration. The observed bands are in excellent agreement with the calculated frequencies of 710.2, 596.8 and 703.2 cm^−1^, respectively. The observation of the Be−H bending mode overwhelmingly confirmed our assignment of HBeCH_2_CN.

The band at 2147.7 cm^−1^ exhibited its D and ^13^C counterparts at 2114.6 and 2075.7 cm^−1^. The B3LYP computed ^12^C/^13^C isotopic frequency ratios for C−N stretching mode of HBeNCCH_2_ was 1.0352, very slightly lower than the observed value of 1.0347. For D counterpart, the calculation result showed that the C−N stretching mode is coupled with CD_2_ symmetric stretching vibration, which is the reason of the large D shift observed in the experiment. The coincident coupled frequency in D substitution is decisive for the confirmation of product HBeNCCH_2_. The Be−H stretching mode was observed at 1661.8 and 2169.0 cm^−1^ for D and ^13^C, and the predicted values were at 1688.5 and 2197.2 cm^−1^, respectively. The calculated intensities of the other infrared bands are all weak in Appendix A.

## 3. Molecular Structures and Bonding

The optimized structures of diberyllium and mono beryllium end-on and insertion complexes are shown in Figure 3, with the end-on products in C_s_ symmetry and the insertion complexes possessing C_3v_ symmetry. The point groups of beryllium products happen to be reverse with the C_3v_ point groups of N-coordination transition-metal complexes and C_s_ structures of transition-metal insertion complexes (except for Mn products) [30,31,32,33,34,35,36,37]. According to the NBO analysis in Appendix A, the electropositive beryllium forms *sp* hybrid orbitals, which make σ bonds with *sp* hybrid orbitals on the C atoms in the insertion complexes. The higher *s* contributions from Be in the C−Be bond lead to the linear structures, different from the higher *d* characters in the carbon−metal bond leading to the bent transition-metal structures. The unpaired spin destinies of triplet state BeNCCH_3_ complex are most located on Be and C atoms (1.105 and 0.555, shown in Appendix A), which cause a bent structure with CCN bond angles of 134.6°.

Unlike the tiny differences in the structures, the diberyllium products are much more exothermic compared with mono beryllium complexes, which may be caused by the formation of the Be−Be real single bonds. The binding energies of the Be–Be bonds in BeBeNCCH_3_ and CNBeBeCH_3_ are calculated to be −27.1 and −77.2 kcal/mol at CCSD(T)/aug-cc-pVTZ level of theory, respectively. The Be–Be bond in CNBeBeCH_3_ is also quantified by the EDA, as shown in Figure 4 and Appendix A. The total interaction energy ΔE_int_ is −73.72 kcal·mol^−1^ between the NCBe and BeCH_3_ fragments, which consist of −65.54 kcal·mol^−1^ of electrostatic energy ΔE_elstat_, −41.61 kcal·mol^−1^ of orbital interaction energy ΔE_orb_, and 33.43 kcal· mol^−1^ of Pauli repulsion energy ΔE_Pauli_. For BeBeNCCH_3_, the Be–Be bond distance is calculated to be 2.077 Å, significantly shorter than that in the free Be_2_ dimer of 2.509 Å. The Be−Be distance is further shortened in CNBeBeCH_3_ with 2.058 Å. The Be–Be distances match the single bond value of 2.05 Å in reported calculations of comparable compounds, and the bond lengths are also closer to that of Be_2_^2+^ than to the neutral Be_2_ molecule [17]. Furthermore, the CN and CH_3_ moieties are both doubly occupied rather than singly occupied in HOMO-2 and HOMO-3 (Appendix A), which also indicates charge transfer from Be_2_ to the bonding fragments. As expected, the beryllium atoms in our diberyllium products carry positive charge and the NC and CH_3_ moieties are negatively charged. According to the NPA charges showed in Figure 3, the Be_2_ moiety carries positive charges of 0.38 and 1.32 for BeBeNCCH_3_ and CNBeBeCH_3_, respectively. More positive charges on the Be_2_ moiety lead to shorter Be−Be bond distances.

The energy decomposition analysis (EDA) in combination with the natural orbital for chemical valence (NOCV) theory has been carried out to further confirm the types of bonding in Table 2. There are two possibilities of interaction types in CNBeBeCH_3_: ionic or neutral interacting fragments. The most suitable fragments to describe the bonding situations yield the smallest amount of the orbital interaction energy (ΔE_orb_), because the least alteration of the electronic charge distribution is required to generate the electronic structure of the molecule. The significantly smaller ΔE_orb_ value of ionic fragments (by 183.7 kcal/mol) convincingly shows that CNBeBeCH_3_ complex can be envisaged as the result of the interaction of Be_2_^2+^ and (NC···CH_3_)^2−^ through a donor–acceptor type of bonding rather than the electron-shared bonding between the neutral ones. The substantial electrostatic energy ΔE_elstat_ (−640.7 kcal/mol) between the ionic interacting fragments contributes much more (75.9%) to the total attraction than the covalent contribution ΔE_orb_ (24.0%). The polarization interaction caused by strong electrostatic interaction is also not neglectable. The corresponding deformation densities Δρ due to the electron transfer between the ionic interacting fragments are visualized in Figure 4. The direction of the charge flow is from the red region to the blue one. The colors also corroborate well with the NPA charges in Figure 3. It is obvious that the orbital interactions in CNBeBeCH_3_ are mainly σ donations of the Lewis base (NC···CH_3_)^2−^ to the vacant σ_u_^+^ orbitals of the Lewis acid Be_2_^2+^.

We also studied the topological analysis of electron density in these complexes to obtain a better understanding of the bonding situation. Figure 5 and Appendix A display the contour plots of Laplacian of electron density (∇^2^ρ(r)) at the molecular plane. The red dotted lines correspond to the areas of charge concentration (∇^2^ρ < 0), while the blue solid lines indicate the areas of charge depletion (∇^2^ρ > 0). The bond critical points identified between two beryllium atoms with negative Laplacian values (∇^2^ρ_cp_ = −0.1176 in BeBeNCCH_3_, −0.1079 in CNBeBeCH_3_) support the formation of a real Be−Be bond, which exhibits covalent character. The negative values of the electronic energy density, E(r), as −0.3154 and −0.3997 in Appendix A, also stand for the interactions with significant covalent character. All Be−N interactions present a positive bond critical point (BCP) value for the Laplacian ∇^2^ρ(r) (∇^2^ρ_cp_ = 0.6479 in BeBeNCCH_3_, 0.7387 in BeNCCH_3_, 0.6078 in CNBeBeCH_3_, and 0.6073 in CNBeCH_3_), denoting their ionic character. The negative electron energy densities at BCP (E(r)= −0.2494 in BeBeNCCH_3_, −0.3440 in BeNCCH_3_, −0.3663 in CNBeCH_3_ and −0.3788 in CNBeBeCH_3_) indicate the covalent character. Thus, the Be−N bonds are covalent polar bonds with a certain degree of both covalent and ionic contributions [38]. The localized orbital locator (LOL) map is also introduced to characterize bond effects in Figure 5 and Appendix A. The LOL values of more than 0.9e (red regions) in Be−Be region of our products are significant larger compared with isolated Be_2_ dimer. The strong electron localization indicates the aforementioned Be−Be covalent bonds. The strength of the Be−Be bonds in our products is also clear from the HOMO molecular orbitals involving the Be_2_ moiety in Figure 5. Coherently, NBO describes the Be−Be bonds as σ bonds with a population of around two electrons (Appendix A). The Wiberg bond index (WBI) for the Be–Be bonds is calculated to be 0.791 in BeBeNCCH_3_ and 0.966 in CNBeBeCH_3_. It is worth noting that the strength of the Be−Be bond in CNBeBeCH_3_ is just slightly weaker than the C–Be and N–Be bonds, the bond orders of which are 0.976 and 0.977, respectively.

For HBeCH_2_NC, the C−C and C–N bond distances are calculated to be 1.449 and 1.153 Å, respectively. After Be−H moiety transferring from CH_2_ to the N-end, the C−C bond is sizably shortened to 1.314 Å while the C−N bond is weakened (WBI from 2.923 to 1.924). The Wiberg bond order gives values of 0.985 and 1.825 for the C−C bonds of HBeCH_2_CN and HBeNCCH_2_, indicating single and double bonds, respectively.

## 4. Reaction Mechanism

The calculated reaction paths including energies of products and transition states at the CCSD(T)//B3LYP−D3/aug−cc−pVTZ level of theory are illustrated in Figure 6. The reactions of beryllium atoms with acetonitrile in excess neon can be divided into three separate paths. For all products both singlet and triplet states are considered; unless noted, the following discussion relates to the lower energy singlet state species.

The calculations indicate that beryllium dimer reacts with acetonitrile on co-deposition to form the initial structure, BeBeNCCH_3_. The diberyllium end-on complex is more stable in the singlet (^1^A’) electronic state by exothermic 13.6 kcal/mol. BeBeNCCH_3_ is also possible to be formed by adding another beryllium atom to the mono beryllium end-on complex in our experiments. The next step is both beryllium atoms bonding to the N atom to form a triangle intermediate (not observed), and an energetic barrier of 15.2 kcal/mol needs to be overcome. It later transfers to another energetically much lower cyclic intermediate via one beryllium atom bonding to the C atom by exothermic 45.9 kcal/mol. The infrared absorptions of this cyclic intermediate are calculated and showed in Appendix A. Unfortunately, its main absorptions (1609.8, 1095.5 and 768.1 cm^−1^) are all close to the extremely strong bands of water or precursor, so we cannot make sure whether it existed in our experiment or not. The last energy barrier of insertion into the C−N bond is pretty high with 67.8 kcal/mol, and the most stable diberyllium product CNBeBeCH_3_ is finally generated.

For the mono beryllium reactions, calculations suggest that the initial end-on complex BeNCCH_3_ is more stable in the triplet state, and the laser power in our experiments is able to initiate this N-coordination process. The CNBeCH_3_ complex is calculated to be the global minimum for the mono beryllium products as 61.8 kcal/mol exothermic. The insertion process occurs in a single step involving the breaking of C−C bond and the formation of a new C−Be bond, which only requires to cross an energy barrier of 25.4 kcal/mol. The insertion reaction is also the primary reaction according to the strongest infrared absorptions of the product “**4**” after full arc irradiation. The C−H bond insertion products of acetonitrile in cryogenic matrix experiment were first observed in our experiment. The C−H insertion reaction proceeded with a high energetic cost of 62.9 kcal/mol, which was apparently supplied by photon energy during full arc photolysis, and in line with the experimental observation of HBeCH_2_CN and HBeNCCH_2_ only after full arc irradiation. The HBeCH_2_CN complex is found to be 35.6 kcal/mol more stable than the reactants. The next step involves a relatively low energy barrier of 8.8 kcal/mol to obtain the more stable product, HBeNCCH_2_, which is exothermic by 37.8 kcal/mol.

## 5. Experimental and Computational Methods

The experimental details in conjunction with matrix infrared spectroscopy investigation have been described previously [39,40,41,42]. Briefly, the Nd:YAG laser fundamental (1064 nm, 10 Hz repetition rate with 10 ns pulse width) was focused onto a rotating beryllium metal target (Alfa Aesar, Haverhill, MA, USA), and the ablated beryllium atoms were co-deposited with the gas mixture of neon and acetonitrile (Sinopharm Chemical Reagent Co., Ltd., Shanghai, China) onto a substrate (CsI) maintained at 4 K using a closed-cycle helium refrigerator (Sumitomo Heavy Industries Model RDK 205D, Tokyo, Japan) for normally 60 min. Matrix samples then were annealed and subjected to LED lights or a high-pressure mercury arc lamp (Philips, 175 W, Beijing, China) with the globe removed, and the products were confirmed by IR spectra recorded on a Bruker 80 V spectrometer in the range from 400 to 4000 cm^−1^ at 0.5 cm^−1^ resolution. The experiments were repeated using CD_3_CN (Sigma-Aldrich, St. Louis, MO, USA) and ^13^CH_3_^13^CN (Cambridge Isotope Laboratories, Cambridge, Britain) samples to further confirm our assignment of the products.

Quantum chemical calculations were carried out using the Gaussian 09 software package [43]. The geometry optimization and frequency calculations were performed using the hybrid B3LYP [44,45] functional including Grimme’s D3-dispersion correction [46] with aug-cc-pVTZ [47,48] basis set employed for all atoms. The second-order vibrational perturbation theory (VPT2) was used to calculate the anharmonic values of the spectral parameters [49,50]. Geometries of products were reoptimized using the more strenuous CCSD method [51] and the single-point energy calculations were separately performed at the CCSD(T) [52,53,54]/aug-cc-pVTZ level with B3LYP-D3-optimized geometries. Bonding analyses were carried out considering Quantum Theory of Atoms in Molecules (AIM) [38], localized orbital locator (LOL) map [55] and Natural Bond Orbital (NBO) [56]. The AIM and LOL analyses were performed using Multiwfn code [57] and the NBO analysis was computed using the NBO 3.1 program. The Multiwfn and VMD [58] programs were applied for plots of HOMO molecular orbitals. The energy decomposition analysis (EDA) [59,60] combined with the natural orbital for chemical valence (NOCV) theory [61,62] was performed using the ADF 2020.101 program packages to analyze the interaction nature of the bonding fragments [63,64]. Uncontracted triple-ζ Slatertype orbital (STO) basis sets plus two sets of polarization functions (TZ2P) [65] were used for all elements. The scalar relativistic effects were included via the zeroth-order regular approximation (ZORA) Hamiltonian [66].

## 6. Conclusions

Beryllium atom reactions with CH_3_CN give six new products. The diberyllium complexes BeBeNCCH_3_ and CNBeBeCH_3_ were found to be 30 kcal/mol more exothermic than the mono beryllium products because of the formation of the Be−Be single bonds. The calculated binding energies of the Be–Be bonds are −27.1 and −77.2 kcal/mol for BeBeNCCH_3_ and CNBeBeCH_3_ at CCSD(T)/aug−cc−pVTZ level of theory, respectively. In spite of the fact that Be is an electron-deficient element, the Be_2_ moiety shows a distinct cationic character and the electron transfer from Be_2_ antibonding orbitals to the bonding fragments strengthens the Be−Be bonds. AIM, LOL and NBO analyses confirm the existence of the covalent Be−Be bonds. EDA-NOCV analysis demonstrates clearly that the CNBeBeCH_3_ complex can be seen as a result of the interaction between Be_2_^2+^ and (H_3_C···NC)^2−^ anions. The bonding interactions between Be and N atoms are identified as covalent polar bonds because of the coexisting ionic and covalent characters from AIM analyses. Lower yields of HBeCH_2_CN and HBeNCCH_2_ have also been observed after full arc irradiation, and the double C=C and C=N bonds are formed after Be−H moiety transferring from CH_2_ to the N-end. Finally, we hope the present study can further extend knowledge of the reactions and bonding investigations of beryllium complexes, and attract further theoretical and experimental research about beryllium chemistry.

## Figures and Tables

**Figure 1 molecules-29-00177-f001:**
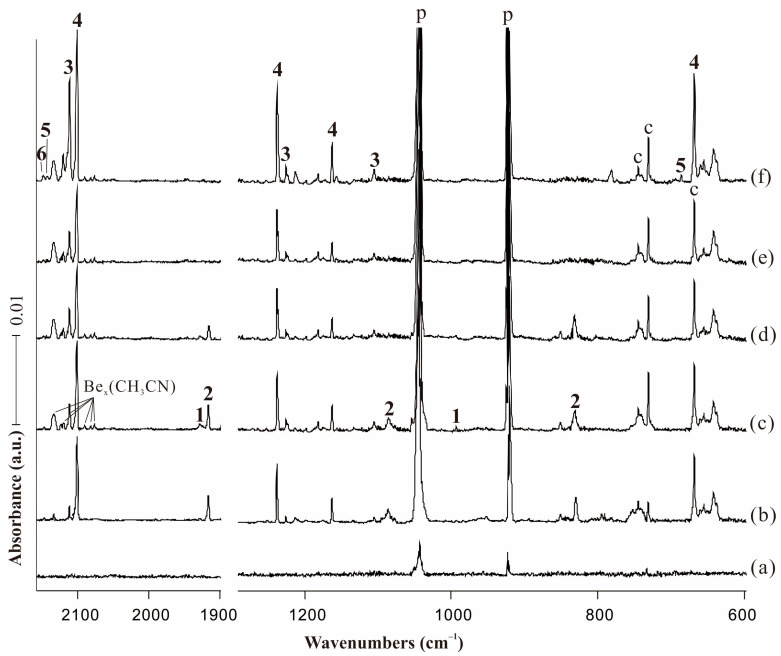
Infrared spectra in the product regions from the reactions of laser-ablated Be atoms with 0.5% CH_3_CN in neon matrix at 4 K. (**a**) Pre-deposition of 0.5% CH_3_CN in neon for 6 min; (**b**) co-deposition for 60 min (weaker laser intensity); (**c**) co-deposition for 60 min (higher laser intensity); (**d**) after annealing at 8 K; (**e**) after visible (λ > 520 nm) irradiation for 6 min; and (**f**) after full arc (λ > 220 nm) irradiation for 6 min. Bands labeled p and c stand for the precursor absorptions and common absorptions in the CH_3_CN matrix spectra. Numbers 1, 2, 3, 4, 5, 6 denote BeBeNCCH_3_, BeNCCH_3_, CNBeBeCH_3_, CNBeCH_3_, HBeCH_2_CN and HBeNCCH_2_, respectively.

**Figure 2 molecules-29-00177-f002:**
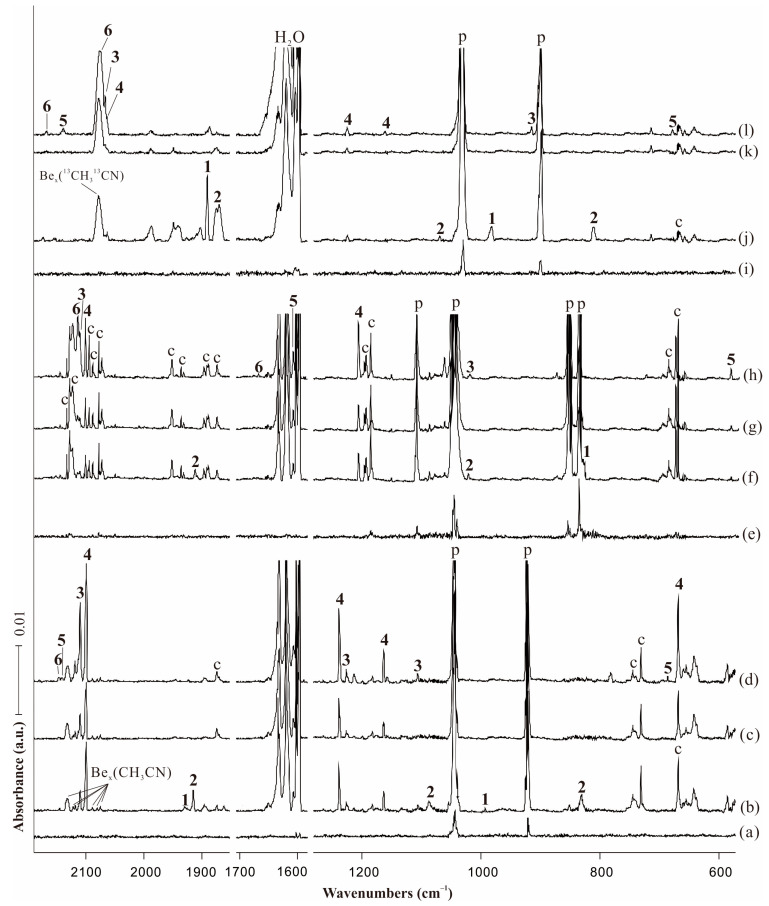
Infrared spectra in the product regions from the reactions of laser-ablated Be atoms with CH_3_CN in neon matrix at 4 K. (**a**) Pre-deposition of 0.5% CH_3_CN in neon for 6 min; (**b**) co-deposition of Be + 0.5% CH_3_CN for 60 min; (**c**) after visible (λ = 520 nm) irradiation; (**d**) after full arc (λ > 220 nm) irradiation for 6 min; (**e**) pre-deposition of 0.75% CD_3_CN in neon for 6 min; (**f**) co-deposition of Be + 0.75% CD_3_CN for 60 min; (**g**) after visible (λ = 520 nm) irradiation; (**h**) after full arc (λ > 220 nm) irradiation for 6 min; (**i**) pre-deposition of 0.75% ^13^CH_3_^13^CN in neon for 6 min; (**j**) co-deposition of Be + 0.75% ^13^CH_3_^13^CN for 60 min; (**k**) after visible (λ = 520 nm) irradiation; and (**l**) after full arc (λ > 220 nm) irradiation for 6 min. Bands labeled p and c stand for the precursor absorptions and common absorptions in the CH_3_CN matrix spectra. Numbers 1, 2, 3, 4, 5, 6 denote BeBeNCCH_3_, BeNCCH_3_, CNBeBeCH_3_, CNBeCH_3_, HBeCH_2_CN and HBeNCCH_2_, respectively.

**Figure 3 molecules-29-00177-f003:**
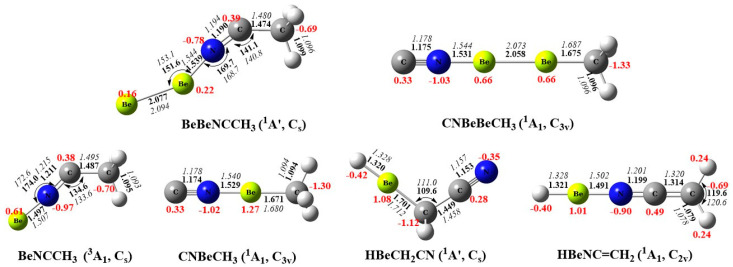
Optimized structures of the reaction products at the B3LYP−D3 (**bold**) and CCSD (*italic*) levels. The aug−cc−pVTZ basis was set for all atoms. Black color: bond length in Å and bond angle in degree; red color: the atomic partial charges calculated by the NBO method.

**Figure 4 molecules-29-00177-f004:**
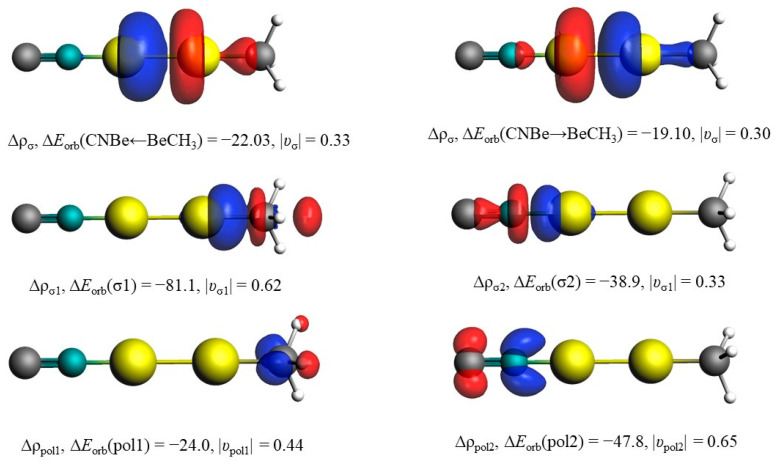
Plots of the deformation densities Δρ. ΔE_orb_ are in kcal mol^−1^. The direction of the charge flow is red → blue.

**Figure 5 molecules-29-00177-f005:**
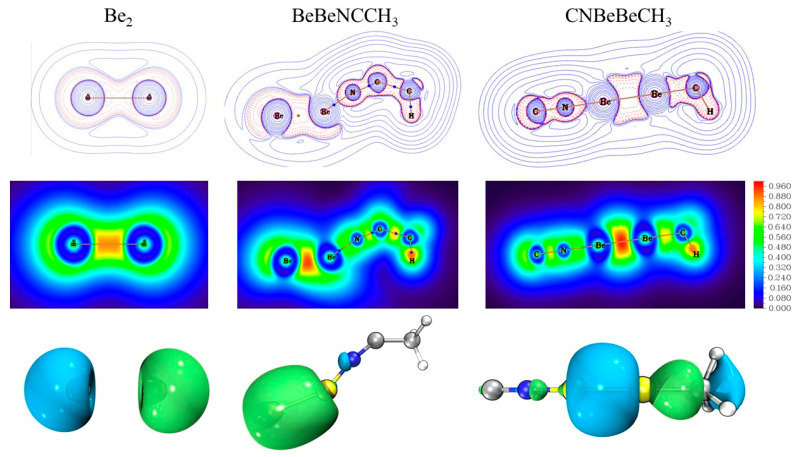
Contour line diagrams of the Laplacian electronic density of the diberyllium products (**first row**). Blue dots stand for BCPs. The blue solid lines and red dotted lines correspond to values of ∇^2^ρ(r) > 0 and ∇^2^ρ(r) < 0, respectively; 2D Localized Orbital Locator (LOL) map of the diberyllium products (**second row**). The HOMO molecular orbitals involving the Be_2_ moiety (**last row).** The length unit is Bohr for Laplacian and LOL map.

**Figure 6 molecules-29-00177-f006:**
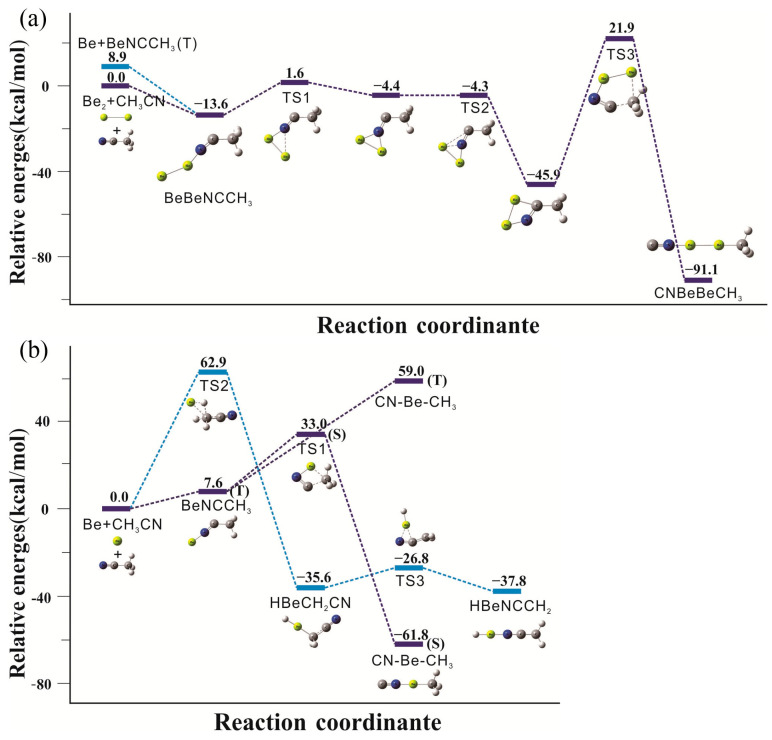
Calculated reaction paths in relation to the reactants [Be/Be_2_ + CH_3_CN]. (**a**) diberyllium reaction paths; (**b**) mono beryllium reaction paths. Relative energies are given in kcal mol^−1^. S and T denote singlet and triplet electronic states, respectively.

**Table 1 molecules-29-00177-t001:** Experimentally observed (in Ne) and calculated vibrational frequencies of products from reactions of beryllium atoms with acetonitrile ^a^.

Be + CH_3_CN		Be + CD_3_CN		Be + ^13^CH_3_^13^CN		Mode
Obs.	Cal.	Obs.	Cal.	Obs.	Cal.	
1. BeBeNCCH_3_						
1927.8	1986.0 (1132)	covered ^b^	1984.3 (1151)	1889.4	1942.7 (1098)	C−N str.
994.1	986.0 (581)	827.0	812.3 (754)	981.2	975.1 (504)	CH_3_ bend
2. BeNCCH_3_						
1915.4	1953.5 (119)	1911.2	1952.1 (124)	1875.4	1915.6 (121)	C−N str.
1082.3	1106.0 (76)	1020.8	1043.4 (77)	1068.7	1091.5 (79)	N−Be str.
831.4	807.0 (71)	covered ^b^	875.8 (82)	811.6	790.0 (67)	C−C str.
3. CNBeBeCH_3_						
2111.0	2157.7 (458)	2111.9	2157.7 (458)	2068.1	2117.7 (458)	N−C str.
1226.1	1249.1 (48)		1037.2 (3)		1237.2 (44)	CH_3_ wag
1105.6	1109.8 (44)		831.2 (56)		1105.9 (39)	Be−Be str.
Covered ^b^	909.5 (185)	1013.4	1004.5 (212)	913.4	902.2 (185)	C−Be str.
4. CNBeCH_3_						
2100.2	2164.8 (436)	2101.4	2165.0 (428)	2064.3	2125.0 (438)	N−C str.
1237.8	1269.2 (160)	1205.0	1217.9 (301)	1223.8	1255.5 (146)	CH_3_ wag
1162.9	1186.4 (147)		951.4 (0)	1160.5	1183.2 (152)	CBeN as. str.
668.3	695.6 (88)		576.0 (83)		590.5 (87)	CBeN bend
5. HBeCH_2_CN						
2141.4	2196.1 (159)	1608.3	1654.6 (121)	2141.8	2195.9 (160)	Be−H str.
685.9	710.2 (139)	579.9	596.8 (96)	680.6	703.2 (141)	Be−H bend
6. HBeNCCH_2_						
2147.7	2204.0 (1309)	2114.6	2171.9 (1129)	2075.7	2197.2 (590)	C−N str.
	2179.9 (287)	1661.8	1688.5 (232)	2169.0	2129.1 (966)	Be−H str.

^a^ All frequencies are in cm^−1^ and the calculated IR intensities are in the parentheses (km mol^−1^). The computed frequencies are given at the B3LYP−D3/aug−cc−pVTZ level of theory. ^b^ Absorptions covered by precursor bands.

**Table 2 molecules-29-00177-t002:** EDA-NOCV results for CNBeBeCH_3_ obtained at the B3LYP/TZ2P/ZORA level of theory.

Energy Term ^a^	Orbital Interaction	Be_2_^2+^ + (H_3_C···NC)^2−^	Be_2_ + H_3_C···NC
Δ*E*_int_	-	−687.9	−252.5
Δ*E*_Pauli_	-	155.7	414.4
Δ*E*_elstat_ ^b^	-	−640.7 (75.9%)	−280.2 (42.0%)
Δ*E*_orb_ ^b^	-	−202.9 (24.0%)	−386.6 (58.0%)
Δ*E*_orb_(σ_1_) ^c^	Be_2_^2+^ ← (CH_3_)^−^ σ-donation	−81.1 (40.0%)	-
Δ*E*_orb_(σ_2_) ^c^	Be_2_^2+^ ← (NC)^−^ σ-donation	−38.9 (19.2%)	-
Δ*E*_orb_(pol_1_) ^c^	H → C polarization	−24.0 (11.8%)	-
Δ*E*_orb_(pol_2_) ^c^	C → N polarization	−47.8 (23.6%)	−26.2 (6.8%)
Δ*E*_orb_(σ_3_) ^c^	Be_2_ → NC σ-backdonation	-	−201.0 (52.0%)
Δ*E*_orb_(σ_4_) ^c^	Be_2_ → CH_3_ σ-backdonation	-	−105.5 (27.3%)
Δ*E*_orb_(σ_5_) ^c^	Be_2_ ← CH_3_ σ-donation	-	−17.7 (4.6%)
Δ*E*_orb_(σ_6_) ^c^	Be_2_ ←CH_3_ σ-donation	-	−24.9 (6.4%)
Δ*E*_orb(rest)_ ^c^	-	−11.1 (5.5%)	−11.3 (2.9%)
Δ*E*_prep_	-	60.3	24.2
Δ*E* (-*D*_e_)	-	−627.6	−228.3

^a^ Energy values are given in kcal/mol. ^b^ The percentage contribution to the total attractive interactions (Δ*E*_elstat_ + Δ*E*_orb_ + Δ*E*_disp_) is given in parentheses. ^c^ The percentage contribution to the total orbital interactions is given in parentheses.

## Data Availability

Data are contained within the article and Appendix A.

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
