# Peer review of "Beryllium Dimer Reactions with Acetonitrile: Formation of Strong Be−Be Bonds"

_molecules, 2023, doi:10.3390/molecules29010177_

Round 1

Reviewer 1 Report

Comments and Suggestions for Authors

The manuscript by Cong et al. deals with synthesis of the Be…NCMe and Be2…NCMe species in 4K solid Ne matrix and their IR spectroscopic and theoretical characterization. The general topic is of interest due to scarce information about the nature and stabilization of the Be2 bond. Meanwhile, the work requires significant revision in accord with comments below before it could be accepted.

1. The most serious problem is associated with the computational method used. First, the B3LYP functional should not be adequate for such calculations, in particular, without the D3 correction. In general, given very small size of the structures, the reasonable approach in this case is optimization at CCSD and the single point calculations at CCSD(T) (or even optimization at CCSD(T) with numerical calculations of gradient and frequencies). Second, the basis set is obviously inadequate. I should suggest at least aug-cc-pVTZ or even higher. By the way, addition of the D3 correction to M06-2X is not good because this functional already includes some part of the medium-range correlation.

2. As I understand, the frequencies were calculated at the harmonic approach. This is not reasonable in this case. The anharmonicity should be calculated.

3. Spectra of the free CH3CN, CD3CN and (13)CH3(13)CCN in the matrix should be provided in Figures 1 and 2 for comparison. The changes which occur in the spectra of acetonitrile upon its interaction with Be and Be2 should be briefly discussed.

4. Table 1. The calculated frequencies for Be + CD3CN and Be + (13)CH3(13)CN should be added for comparison with experimental values.

5. Page 6, line 190. “The optimized structures … are strongly similar…” Certainly not. The structures are different because the species are different.

6. The EDA-NOCV analysis was performed for the interaction between Be2 and NCMe. But what is much more interesting is the nature of the Be…Be bond. The energy decomposition analysis should be performed and discussed for this bond as well.

7. The EDA-NOCV analysis was done for CNBeBeCH3 which was decomposed to Be2 and NCMe (or Be2(2+) and NCMe(2-)). Honestly, I cannot understand the meaning of the results obtained. Such a decomposition describes not only the Be…NCMe bonding but also the cleavage the C-C bond in NCMe (because CNBeBeCH3 is an insertion product). The resulting values should be a mixture of properties of the Be-N, Be-C and CC bonds which is fully uninformative. If the authors want to apply this analysis, they should do it separately for each bond (Be-N and Be-C). Also, the same analysis should be performed for BeBeNCMe.

8. Page 8. Please, indicate units for the Laplacian and energy densities in the text.

9. Page 8. “electronic energy density”. Is it a total energy density?

10. Page 8. “All Be-N interaction present a positive … Laplacian, denoting their ionic character. However, the negative electron energy density … indicate the covalent character” So, ionic or covalent? I think, this bond has a certain degree of both covalent and ionic contributions, as any covalent polar bond.

11. The AIM properties should be calculated and provided for all structures under consideration. The electron density values at BCPs should also be provided.

12. Please, add a Table with the AIM parameters.

13. The interaction and binding energies for the formation of the Be…Be bonds in 1 and 3 should be calculated.

14. Figure 5. Please, clearly identify the molecules in this figure.

15. Figure 6 (a). Why not to start this pathway with the formation of BeNCMe which then interacts with another Be atom to give BeBeNCMe?

16. Figure 6 (a). The cyclic intermediate (the energy of -49.7 kcal/mol) is the second more stable species, and its conversion requires very high activation barrier. In this case, this intermediate should be efficiently accumulated and be visible in the IR spectrum. Please, search for it.

17. In 1 and 3, the Be-Be bond may be easily broken. What about the products of such a process? Could they be detected experimentally?

18. The recent work by West (Nature Synthesis, 2023, 2, 696) should be cited.

Comments on the Quality of English Language

Minor editing of English language required

Author Response

My reply to reviewers is attached

Reviewer 2 Report

Comments and Suggestions for Authors

The manuscript studies the effect del Acetonitrile as a good electron donor based on the previous research of reactions with metal atoms, and analyze that could be a good candidate for stabilizing the Be2 dimer. The authors, investigated the reactions of beryllium atoms with acetonitrile by means of matrix-isolation infrared spectroscopy and theoretical calculations, in order to further supplement the reactions of alkali metal with acetonitrile and search for stable diberyllium complexes that might be formed. Six different products were spectroscopic identified in solid neon including two complexes possessing Be−Be single bonds, and three related reaction paths were presented.

Before its approval, I believe that the authors should address the following questions:

1.              How were each of the IR signals in Figure 1 assigned for each of the species proposed to be present in the experiment: species 1-6?

2.              The authors, in addition to IR, have other experimental evidence that supports the assignments discussed in IR, for example mass spectroscopy and even NMR of 1H and 13C since the displacements should be affected by the presence of a metal such as Be.

Author Response

My reply to reviewers is attached. 

Round 2

Reviewer 1 Report

Comments and Suggestions for Authors

In the revised manuscript, most of my concerns were addressed. However, there is one important issue to be resolved. As indicated in the reply to my question nº 7 and also in Conclusions, the formation of CNBeBeCH3 should be considered preferrably as an interaction between ions Be2+ and NCMe(2-) rather than as an interaction between neutral species Be2 and NCMe. This conclusion was based on the results of the EDA-NOCV analysis. In fact, the situation is just opposite. The homolytic dissociation energy of CNBeBeCH3 is much lower than the heterolytic one (228.3 vs. 627.6 kcal/mol, Table 2). This means that the formation of CNBeBeCH3 should also be considered as a result of the interaction between neutral Be2 and NCMe.

Just one example: should the formation of fluor F2 be considered as F + F or as F- + F+? The interaction energy of formation of F2 from the ions is incomparably lower (more negative) than that from the neutral atoms. However, no one says that F2 preferably "can be seen as a result of the interaction between" F- and F+.

Once this issue is resolved, the manuscript is recommended for publication.

Comments on the Quality of English Language

Minor editing of English language required
